# MATMAMBA: A MATRYOSHKA STATE SPACE MODEL

## ABSTRACT

State Space Models (SSMs) like Mamba2 are a promising alternative to Transformers, with faster theoretical training and inference times – especially for long context lengths. Recent work on Matryoshka Representation Learning – and its application to Transformer backbones in works like MatFormer – showed how to introduce nested granularities of smaller submodels in one universal elastic model. In this work, we present MatMamba: a state space model which combines Matryoshka-style learning with Mamba2, by modifying the block to contain nested dimensions to enable joint training and adaptive inference. MatMamba allows for efficient and adaptive deployment across various model sizes. We train a single large MatMamba model and are able to get a number of smaller nested models for free – while maintaining or improving upon the performance of a baseline smaller model trained from scratch. We train language and image models at a variety of parameter sizes from 35M to 1.4B. Our results on ImageNet and FineWeb show that MatMamba models scale comparably to Transformers, while having more efficient inference characteristics. This makes MatMamba a practically viable option for deploying large-scale models in an elastic way based on the available inference compute.

## 1 INTRODUCTION

Deep learning practitioners often train different sizes of the same kind of model to facilitate deployment in a variety of ranges of available inference compute. For example, the Llama 3.2 (Dubey et al., 2024) series has 1B, 3B, 11B, and 90B variations. These models are extremely powerful individually – but due to independent training do not necessarily share the same metric space – a property which can be extremely useful for inference applications like speculative decoding (Leviathan et al., 2023), hybrid cloud-edge inference, or just general input or compute adaptive processing. Moreover, because training these models is expensive, we typically see only a few chosen sizes trained. This is not desirable in situations where the deployment setup can optimally support an intermediate model (e.g. a 2B model), but has to settle for the less accurate 1B model instead.

Techniques like model compression and distillation aim to address these issues, but require additional training (for which data may not be available), and can sometimes drop accuracy (Jaiswal et al., 2023). Thus, methods that offer adaptive inference out of the box at intermediate granularities are extremely useful. This has been explored for Transformers (Devvrit et al., 2023; Cai et al., 2024b) and ConvNets (Yu & Huang, 2019; Cai et al., 2019). The core focus of this work is to try to enable out of the box adaptive inference in a newer architecture: Mamba2 (Dao & Gu, 2024).

State Space Models like Mamba2 (Dao & Gu, 2024) and a number of other related newer architectures (see Section 2) have shown tremendous potential as they try to improve on the efficiency of Transformers, while maintaining their potency as accurate and general sequence processing architectures. Mamba2 has comparable scaling properties to Transformers, while being significantly faster at longer context lengths.

In this work, we introduce MatMamba, a nested Matryoshka structure (Kusupati et al., 2022) within a Mamba2 block (Dao & Gu, 2024). MatMamba enables the extraction of hundreds of nested submodels from the same set of weights, without requiring any additional training during deployment. MatMamba is a general-purpose sequence processing architecture that can be applied to any type of model (encoder/decoder), modality (language/vision/sound/actions), loss function, or learning algorithm compatible with a Transformer or Mamba2 layer.

The philosophically closest work to MatMamba is MatFormer (Devvrit et al., 2023) – which imposes a nested structure on the FFN block in a Transformer layer. We use the same concept to impose a nested structure on any learnable parameter in a Mamba2 block that depends upon the hidden dimensionality of the block. Formally, a MatMamba block consists of a nested combination of $g$ Mamba2 blocks $M_i$, such that $M_1 \subset M_2 \subset ... \subset M_g$, where $M_i \subset M_j$ means that all the parameters of a sub-block $M_i$ are present in $M_j$. We train the model using $g$ forward passes with gradient accumulation followed by a single backward pass for parameter updates (see Figure 1).

By jointly training all $g$ granularities, the smallest sub-blocks are incentivized to represent the most important information, like in Matryoshka Representation Learning (Kusupati et al., 2022). We can now use any of the $g$ nested sub-blocks $M_i$ flexibly. Additionally, we can flexibly slice the block along *any* dimensionality (even beyond the $g$ explicitly optimized granularities). Using Mix'n'Match (Section 3.4), we can perform this operation over multiple layers at varying granularities to flexibly extract a combinatorially large number of models from the single larger model. We observe that these extracted models preserve the metric space of the larger model, and are accurate across a variety of tested tasks – effectively allowing us to choose a tradeoff between model performance and compute.

We train MatMamba-based vision models (MatMamba-Vision), and find that: (a) MatMamba-Vision models scale as well as baseline Mamba2 based models at all $g = 4$ granularities; (b) Using Mix'n'Match, we can flexibly extract submodels between the explicitly optimized granularities. The submodels span (and sometimes exceed) the pareto optimal accuracy-vs-compute curve; (c) MatMamba-Vision models are significantly faster at higher resolutions than ViTs, making them promising candidates for long-form and high resolution visual tasks, while enabling adaptive visual processing with the nested submodels (see Section 4.1.1).

Furthermore, MatMamba-Vision models can act as elastic image encoders for adaptive image retrieval. We can encode visual datasets with the largest model, and because the smaller submodels share its metric space, we can use them as query encoders, needing drastically lower compute with minimal loss in accuracy (see Section 4.1.2).

We also train MatMamba-based decoder language models (MatMamba-LM) at various sizes from 130M-1.4B parameters, and at $g = 4$ granularities. We make similar observations here too, that MatMamba-LM models scale as well as Mamba2 baselines with the same architecture for all nested granularities. We also observe interesting homogenous scaling behaviour between the nested granularities for different models (see Section 4.2).

Through MatMamba, for the first time, we bring together the adaptivity of Matryoshka-style learning and the efficiency of state space models (SSMs) like Mamba2 (Dao & Gu, 2024).

**We make the following research contributions:**

1. We introduce MatMamba, which imposes a nested Matryoshka structure on a Mamba2 state space model. We jointly optimize all nested granularities to train a single elastic model.

2. We show that MatMamba models scale as well as the baseline Mamba2 models for a variety of model sizes from 35M-1.4B parameters on language and vision tasks.

3. Using Mix'n'Match with MatMamba allows the flexible extraction of hundreds of submodels to perform adaptive inference. These submodels preserve the metric space of the original model.

4. MatMamba-Vision models are comparably accurate and significantly faster at higher resolutions than ViTs, making them well suited for long-form/high resolution and adaptive visual processing.

## 2   RELATED WORK

The ever growing demand of AI models across various accuracy and resource constraints makes it infeasible to train a different model for each use case. Instead, these adaptive deployment needs are often solved through introducing elasticity in models (Kusupati, 2024). Work on slimmable networks (Yu et al., 2018; Yu & Huang, 2019) and once-for-all networks (Cai et al., 2019) brought the idea of training multiple submodels present within one universal model. Nested dropout (Rippel et al., 2014) generalizes this idea to learn ordered representations which further extended to enable elasticity at each dense vector embeddings through Matryoshka Representation Learning

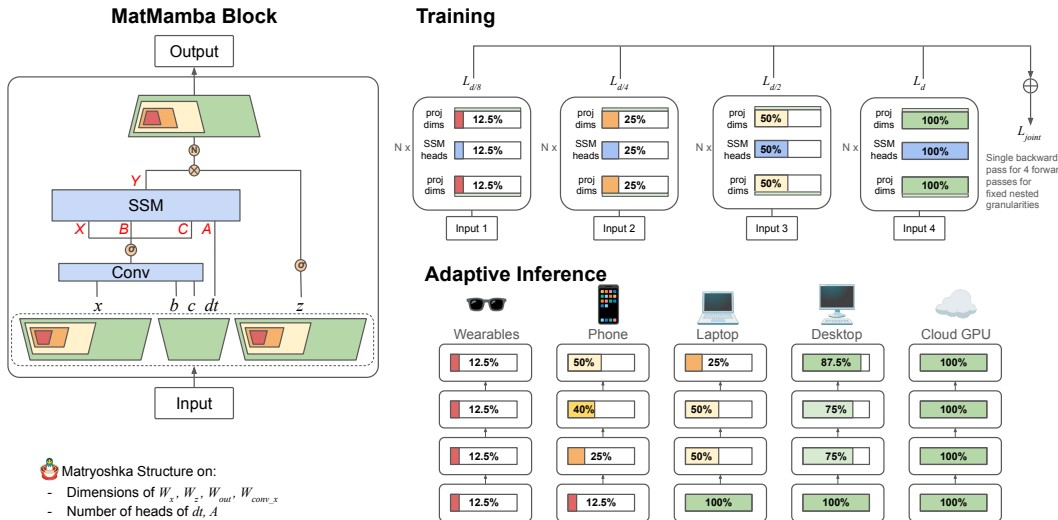

Figure 1: MatMamba introduces a nested Matryoshka (Kusupati et al., 2022) structure in a Mamba2 (Dao & Gu, 2024) block. We jointly train a few chosen granularities to get a single model from which we can flexibly extract a large number of nested submodels for adaptive inference based on the available deployment compute.

(MRL) (Kusupati et al., 2022). MRL simplifies the training process to induce elasticity with a small set of nested granularities (hence the name Matryoshka), exponentially separated in size, all optimized with the same target loss function as the full vector. MRL further smoothly interpolates to the granularities not seen during training, thus allowing for complete elasticity to extract sub-vectors based on the requirements.

Matryoshka information packing and learning has been widely adopted in bringing adaptivity not only in output space, but also in input (Beyer et al., 2023) and model weights (Devvrit et al., 2023; Cai et al., 2024b; Valipour et al., 2023). MatFormer (Devvrit et al., 2023) is a direct translation of MRL to every hidden activation vector of a MLP sub-block within a Transformer layer (Vaswani et al., 2017). MatFormer showed scaling trends similar to Transformer, while also providing the capability to adaptively extract submodels that fall on the accuracy-vs-compute pareto curve. More recent works (Cai et al., 2024b; Jain et al., 2024) developed dynamic routing on top of the conditional computation enabled by MatFormer to realize performance gains in deployment. Further, matryoshka packing was also used for flexible tokenization (Cai et al., 2024a; Hu et al., 2024) as well as diffusion models (Gu et al., 2023).

Transformers (Vaswani et al., 2017) have been fundamental sequence processing blocks in neural networks for the past few years. There has been a recent wave of work on efficient sequence processing architectures that aim to be faster and equally performant alternatives to Transformers. Mamba (Gu & Dao, 2023) and Mamba2 (Dao & Gu, 2024) are the most relevant to this work, with other very closely related works like Linear Attention (Katharopoulos et al., 2020), Test-time training (Sun et al., 2024), RWKV (Peng et al., 2023), Griffin (De et al., 2024), Jamba (Lieber et al., 2024), xLSTM (Beck et al., 2024), HGRN2 (Qin et al., 2024), RetNet (Sun et al., 2023), RecurrentGemma (Botev et al., 2024). Waleffe et al. (2024) present a detailed study of how to train large-scale Mamba-based language models. Works like MambaVision (Hatamizadeh & Kautz, 2024), MambaND (Li et al., 2024b), Vision Mamba (Zhu et al., 2024), VideoMamba (Li et al., 2024a), and Sonic (CartesiaAI, 2024) have all shown how a Mamba layer can process visual data and other modalities. Liu et al. (2024) present a detailed survey of Mamba-based vision models.

# 3 MATMAMBA

## 3.1 MAMBA2 PRELIMINARIES

MatMamba is based on Mamba2. We make simple modifications to the Mamba2 block to impose the Matryoshka structure. A detailed description of the internals of Mamba2 can be found in the original paper Dao & Gu (2024). However for the purposes of this work, we treat the Mamba2 block as a combination of an input linear projection ($W_{in}$, which can be broken down into $W_z$, $W_x$, $W_B$, $W_C$, $W_{dt}$), a causal 1D convolution layer with kernel size 4 (with weights that are a concatenation of $W_{conv_x}$, $W_{conv_B}$, and $W_{conv_C}$ applied in groups), a chunk + selective scan operation ($SSM$), and an output projection layer ($W_{out}$). Similar to a Transformer, this block takes in an $(b, l, d)$ shaped tensor – $b$ is batch size, $l$ is sequence length, and $d$ is the dimensionality – and produces a $(b, l, d)$ shaped output after a sequence transformation. For an input tensor $u$, the Mamba2 block $M(u)$ consists of the following steps:

$$XBC(u) = \sigma(Conv(W_{conv_x} \frown W_{conv_B} \frown W_{conv_C}, W_x.u \frown W_B.u \frown W_C.u)) \quad (1)$$

$$Y(u) = SSM(XBC(u), W_{dt}.u, A, D) \quad (2)$$

$$M(u) = Norm(Y(u).\sigma(W_z.u)).W_{out}^T \quad (3)$$

where $\frown$ is the concatenation operation, $Conv(k, s)$ applies a 1-D causal convolution with weights $k$ (applied in $len(k)$ groups) on a sequence $s$, and $A$ and $D$ are learnable SSM parameters. $\sigma$ is a nonlinearity which we set to SiLU (Elfwing et al., 2018), and *Norm* is a layer norm function which we set to RMSNorm (Zhang & Sennrich, 2019).

## 3.2 MATMAMBA BLOCK

A MatMamba block also has both input and output shapes as $(b, l, d)$. It is defined as a nested combination of $g$ Mamba2 blocks $M_i$, such that $M_1 \subset M_2 \subset ... \subset M_g$, where $M_i \subset M_j$ means that all the parameters of a sub-block $M_i$ are present in $M_j$. Works like MatFormer (Devvrit et al., 2023), OFA (Cai et al., 2019), and Flextron (Cai et al., 2024b) all share similar designs in which the largest model $M_g$ is the single universal base model from which numerous smaller submodels $M_i$ can be flexibly extracted. In MatMamba, we impose the nested structure along the *dimensions* of the model parameters. Specifically for a sub-block $M_i$ with expansion factor $e = \frac{d_{inner}}{d_{model}}$, we choose a Matryoshka dimension $m_i$, such that $0 < m_i < d_{model}$, which results in an inner slice dimension $d_i = e \times m_i$ and number of heads $h_i = \frac{d_i}{d_{head}}$, subject to $d_i \mod d_{head} = 0$. For example, parameters like $W_x$ have a shape of $(d_{inner}, d_{model})$. For the $M_i$ sub-block, it will become $W_x[0 : d_i]$ by slicing it along the $d_{inner}$ dimension. Similarly for parameters like A which have a shape of $(n_{heads})$, it will become $A[0 : h_i]$. Concretely, the MatMamba block $M_i(u)$ when applied to an input tensor $u$ is these steps:

$$XBC_i(u) = \sigma(Conv(W_{conv_x}[0 : d_i] \frown W_{conv_B} \frown W_{conv_C}, W_x[0 : d_i].u \frown W_B.u \frown W_C.u)) \quad (4)$$

$$Y_i(u) = SSM(XBC_i(u), W_{dt}[0 : h_i].u, A[0 : h_i], D[0 : h_i]) \quad (5)$$

$$M_i(u) = Norm(Y_i(u).\sigma(W_z[0 : d_i].u)).W_{out}[0 : d_i]^T \quad (6)$$

In practice, $W_z$, $W_x$, $W_B$, $W_C$, and $W_{dt}$ are implemented as a single input projection layer with tensor parallelism, with appropriate rearranging of dimensions depending on $m_i$. Figure 1 illustrates the MatMamba block. We also provide PyTorch-style pseudocode for the block in Appendix A, to provide a clearer understanding of our implementation.

Compared to MatFormer (Devvrit et al., 2023), where the Matryoshka structure is only applied on the MLP subblock of the Transformer block, MatMamba applies nesting to the entire block wherever the inner dimension plays a role. This leads to a nearly linear reduction in total parameter count (and also a nearly linear reduction in flop count due to the nature of Mamba2). Also, typically $> 95\%$ of the parameter count in a MatMamba block is in the input and output projections, which can be converted into nested layers while maintaining the well understood systems characteristics of projection layers. See Appendix A for a detailed example of parameter count reduction.

We can stack $L$ such MatMamba blocks to create a MatMamba model. For a given $m_i$ and nested blocks $M_1 \subset M_2 \subset ... \subset M_g$, we can create a MatMamba model $f_i$ with $L$ layers, and $g$ nested models $f_1 \subset f_2 \subset ... \subset f_g$. Each $f_i$ is formed by stacking $M_i$ $L$ times. Like Mamba2, the Mat-Mamba backbone is a general purpose sequence processing architecture, which with an appropriate tokenizer and output head can process a variety of modalities.

### 3.3 TRAINING

To train a model comprised of MatMamba blocks for $g$ chosen granularities, we perform $g$ forward passes to calculate a joint loss function. For an input $x$, model $f$, target $y$ and loss function $\mathcal{L}$:

$$\mathcal{L}_{joint}(x, y) = \sum_{i=1}^{g} \lambda_i . \mathcal{L}(f_i(x), y) \tag{7}$$

where $\lambda_i$ is the weight of the $i$-th nested submodel's loss. In this work, we train $g = 4$ nested submodels with a uniform $\lambda_i = 1/g = 0.25$ for each submodel. As shown in Figure 1, during each forward pass, we accumulate gradients. The parameter update is done with a single backward pass. During the whole process, the model and the weights are the same, thereby also making memory usage the same as a regular Mamba2 block. In this work, we train MatMamba models with $g = 4$ nested granularities, with the corresponding list of $m_i$'s being $[d_{model}, d_{model}/2, d_{model}/4, d_{model}/8]$, i.e. a halving of dimensionality for every sub-model. Like MatFormer (Devvrit et al., 2023) and Flextron (Cai et al., 2024b), we note that it is also possible to finetune an existing pretrained model to produce a nested structure. However, in this work, we focus on training from scratch to study the scaling characteristics of MatMamba models.

### 3.4 MIX'N'MATCH

We can apply the Mix'n'Match strategy from MatFormer (Devvrit et al., 2023) to flexibly extract *any* submodel from MatMamba for inference. Concretely, for a model $f$ with $L$ layers, we need to choose a dimensionality $m_i$ at each layer $i$. Note that $m_i$ can be either one of the explicitly optimized $g$ granularites (e.g. picking from one of [1024, 512, 256, 128] from a 135M-MatMamba-Vision model, see section 4.1), or we can choose interpolated dimensionalities that were not explicitly optimized for (e.g. picking any random valid value like 768 or 384 that was not explicitly trained). For instance, we could choose $m_1 = 256$ (25% size) in layer 1, $m_2 = 1024$ (100% size) in layer 2, $m_3 = 768$ (75% size) in layer 3, and so on. The only constraint on $m_i$ in MatMamba is that it needs to lead to an integer number of heads, or that $(e \times m_i) \mod d_{head} = 0$, where $e = \frac{d_{inner}}{d_{model}}$. This leads to a combinatorially large number of possible submodels (beyond the $g$ explicitly optimized granularities) that can be flexibly extracted – all from the same set of base model weights – as shown in Figure 1. Due to the Matryoshka structure, the first few dimensions (that are shared among all the nested submodels) are incentivized to learn the strongest representations.

### 3.5 ELASTIC INFERENCE

When deploying a MatMamba model for inference, we typically need to store the single universal model $f_g$ in memory. If compute is not constrained (or if the inference workload is predictable), then we can use the full model to get the most accurate results. However, depending on dynamic constraints (e.g. available inference compute, energy usage, system load, desired accuracy etc.), we can perform a forward pass on a chosen slice of the network on the fly.

There are exciting possibilities like combining cloud and edge inference – we could store a smaller model $f_i$ on the edge device and when necessary, use the larger model $f_j$ on the cloud, or using a smaller model to act as a draft model for speculative decoding (Leviathan et al., 2023) with a larger verifier model. We could also potentially do input-adaptive sub-model selection (e.g. use a larger model for a more difficult input). All of these are possible only because MatMamba has a consistent and nested Matryoshka structure, in which all the sub-models share the same metric space.

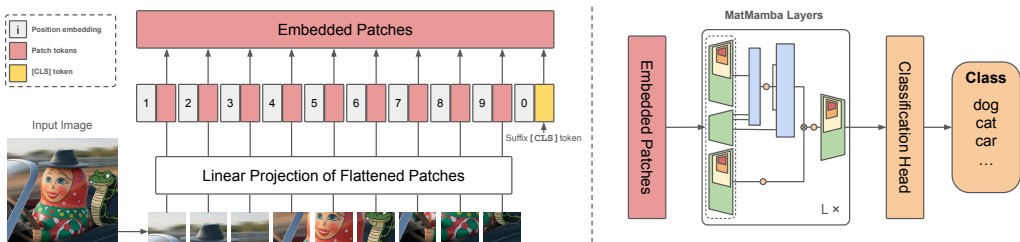

Figure 2: MatMamba layers for vision tasks. Similar to a ViT (Dosovitskiy, 2020), we convert an image into a tensor of embedded patches. Because of the causal nature of the Mamba2 block, we suffix the [CLS] token. We intentionally keep the design simple to better study the properties of the MatMamba block.

## 4 EXPERIMENTS

In this section, we demonstrate the effectiveness of MatMamba-based models across two modalities: vision (**MatMamba-Vision**) and language (**MatMamba-LM**). For vision, we show results for image classification (Section 4.1.1) and adaptive image retrieval (Section 4.1.2). For language, we train decoder language models (Section 4.2). We train models at a variety of scales from 35M to 1.4B parameters. For a fair comparison, we also independently train baseline Mamba2 models which have the same architecture as the submodels of each MatMamba granularity. Please note that *we do not aim to achieve state-of-the-art results* in this work on either language or vision for the chosen model sizes. We instead focus on properties like nested structure consistency, parameter reduction, inference speedups/memory usage for submodels, and scaling of simple networks built using the MatMamba block.

### 4.1 MATMAMBA-VISION

MatMamba-Vision (Figure 2) contains a patch embedding followed by $L$ MatMamba blocks with a unidirectional SSM scan. One crucial design choice we make is to use the [CLS] token as a suffix instead of the conventional prefix. This allows it to attend to information from the entire sequence. We find that this simple architecture works effectively on both image classification and adaptive retrieval. We train two model variations (35M with $d_{model} = 512$ and 135M with $d_{model} = 1024$, see Table 1) with patch size 16 and $L = 20$ layers on ImageNet-1k Deng et al. (2009) which has 1.28M training images and 50k validation images. Compared to other recent work on SSM's for vision tasks like MambaVision (Hatamizadeh & Kautz, 2024), MambaND (Li et al., 2024b), and Vision Mamba (Zhu et al., 2024) – all of which have major design changes on top of Mamba layers like bidirectional scan with additional projections, varying order of scans, or combining SSM layers with attention and convolution layers – we keep our network architecture as simple as possible.

Table 1: Base model architectures for MatMamba-Vision (35M and 135M) with the explicitly optimized submodels for $g = 4$ nested granularities.

| Base Model | Layers | $m_i$ | $h_i$ | Parameters Patch embed | MatMamba Layers |
|---|---|---|---|---|---|
| 135M-1024D | 20 | 1024 | 32 | 787,456 | 132,739,840 |
| | 20 | 512 | 16 | 787,456 | 69,004,160 |
| | 20 | 256 | 8 | 787,456 | 37,136,320 |
| | 20 | 128 | 4 | 787,456 | 21,202,400 |
| 35M-512D | 20 | 512 | 32 | 393,728 | 34,927,360 |
| | 20 | 256 | 16 | 393,728 | 18,787,200 |
| | 20 | 128 | 8 | 393,728 | 10,717,120 |
| | 20 | 64 | 4 | 393,728 | 6,682,080 |

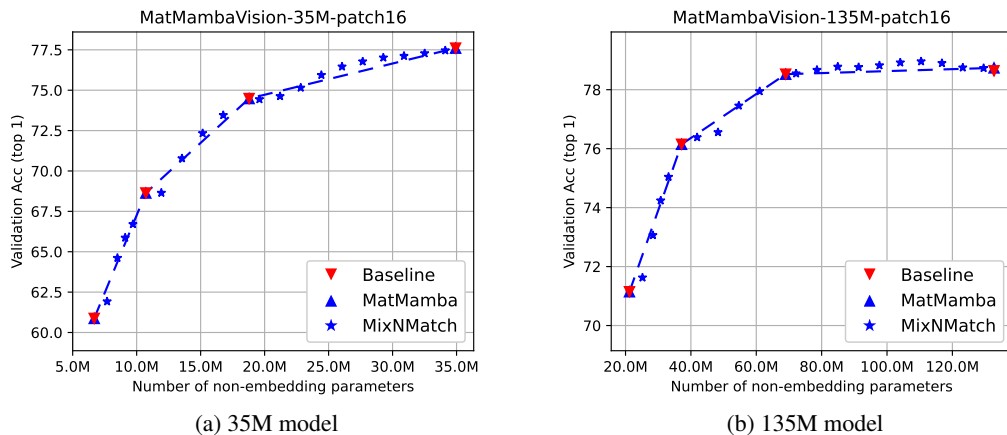

(a) 35M model           (b) 135M model

Figure 3: ImageNet-1K Classification: MatMamba-Vision is as accurate as explicitly trained baselines across various constraints while also spanning the accuracy-vs-compute pareto optimal curve through mix'n'match submodels.

We use FFCV (Leclerc et al., 2023) dataloaders for efficient training. We apply augmentations like RandAug (Cubuk et al., 2020), Random Erasing (Zhong et al., 2020), Mixup (Zhang, 2017), Cutmix (Yun et al., 2019), and a number of other settings following DEiT-3 (Touvron et al., 2022), AugReg (Steiner et al., 2021), and Better ViT Baselines (Beyer et al., 2022). The exact detailed experimental settings can be seen in Appendix A.

### 4.1.1 IMAGE CLASSIFICATION

In Figure 3, we see that for both the 35M and 135M MatMamba-Vision models, the explicitly optimized submodels closely match the 4 independently trained baseline models with the same architecture as the nested submodel. However, instead of needing four separate models, we can get all levels of performance/parameter counts flexibly in a single model.

**Adaptive Inference using Mix'n'Match:** Additionally (Figure 3), using Mix'n'Match at a variety of combined granularities yields models that smoothly interpolate (and sometimes exceed) the accuracy on the line joining the explicitly optimized granularities. This points towards powerful adaptivity, because we can extract a combinatorially large number of submodels along the accuracy-

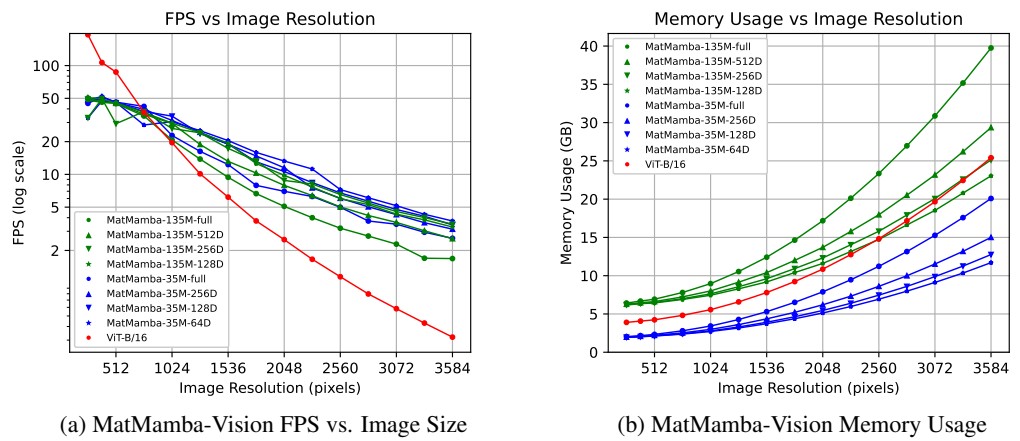

(a) MatMamba-Vision FPS vs. Image Size      (b) MatMamba-Vision Memory Usage

Figure 4: Inference speed and memory usage for batch size 1 on an H100 for nested MatMamba-Vision models and a ViT baseline. At larger resolutions, the characteristics of MatMamba are better.

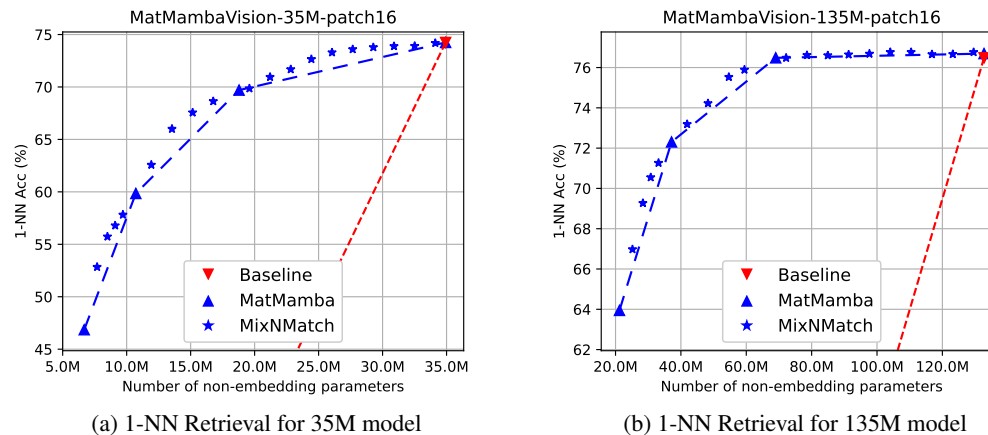

(a) 1-NN Retrieval for 35M model      (b) 1-NN Retrieval for 135M model

Figure 5: Adaptive Image Retrieval on ImageNet-1K: Submodels obtained from the largest MatMamba-Vision model preserve the metric space of embeddings resulting in accurate and adaptive query processing at scale while baseline struggles to work across models without distillation.

compute curve. We can optimize submodel selection for deployment constraints flexibly, all while only using the weights of a single nested universal model.

**Inference Speeds at Higher Resolutions:** In Figure 4, we also study the inference speed tradeoffs of nested granularities of MatMamba-Vision models when compared with each other and a ViT-B/16 model. We find that at or below 512px resolution, the sequence length is low enough for the ViT to be the fastest model (due to GPU parallelism and optimizations like FlashAttention). However, as we increase the resolution to 1024px and beyond, Mamba-style models start outperforming ViT in both throughput and latency. We also study inference memory usage, and find that MatMamba-Vision scales slightly better than an optimized ViT-B/16 as the resolution increases. Both of these observations offer promising evidence that MatMamba-based models can be suitable for processing longer sequences of visual data at higher resolutions on a single accelerator (as opposed to scaling context length in Transformers using methods like RingAttention Liu et al. (2023) which needs multiple interconnected accelerators for a single forward pass at long sequence lengths).

### 4.1.2 ADAPTIVE IMAGE RETRIEVAL

Image retrieval aims to locate semantically similar images using representations generated by a pretrained encoder (Chen et al., 2022). The standard method involves encoding both database and query images with the same encoder and then performing nearest neighbor retrieval. While using a powerful encoder for database images is feasible, the query encoder must be efficient for real-time applications. Moreover, query encoding scenarios can vary, such as on-device versus cloud processing and varying query load and complexity. Existing solutions with fixed encoders often compromise accuracy or cost in different settings.

Due to its flexibility, MatMamba-Vision is a promising candidate for query encoding. However, retrieval also requires that submodels maintain distance relationships between fixed database (encoded with a larger encoder) and query embeddings across various granularities. Using smaller baseline Mamba2 models solely for query encoding can lead to significant distance preservation issues and poor retrieval accuracy (as illustrated in Figure 5).

We evaluated both the baseline and MatMamba-Vision encoders on ImageNet-1K for image retrieval at 35M and 135M parameter scales. Using the [CLS] token representation, we calculated 1-nearest neighbor (NN) accuracy. Figure 5 demonstrates that submodels extracted from MatMamba can effectively preserve distances and offer greater flexibility. For example, MatMamba-Vision-135M can reduce compute cost by 55% with a minimal accuracy loss of less than 0.5%. While causal models with suffix [CLS] token might not be as accurate as bi-directional encoders for retrieval, this is a promising start towards better long-context encoders while enabling adaptive query processing.

Table 2: Base model architectures for MatMamba-LM

| Base Model | Layers | $d_{model}$ | $d_{head}$ | Embed params | Non-embed params | Tokens |
|---|---|---|---|---|---|---|
| 130M | 24 | 768 | 24 | 38,615,040 | 90,368,448 | 62.9B |
| 370M | 48 | 1024 | 32 | 51,486,720 | 316,851,712 | 125.8B |
| 790M | 48 | 1536 | 48 | 77,230,080 | 702,918,912 | 125.8B |
| 1.4B | 48 | 2048 | 64 | 102,973,440 | 1,240,767,488 | 251.6B |

## 4.2 MATMAMBA-LM

We train decoder language models using the MatMamba block (MatMamba-LM). The models closely follow the training procedure and hyperparameters of `llm.c` (Karpathy, 2024). We use the GPT-2 (Radford et al., 2019) tokenizer with a padded vocabulary size of 50,280. We use the FineWeb (Penedo et al., 2024) dataset to train all models. We train 4 separate models (with base model parameter sizes 130M, 370M, 790M, and 1.4B). For each of these base models, we optimize $g = 4$ nested granularities $[d_{model}, d_{model}/2, d_{model}/4, d_{model}/8]$. For baselines, we train vanilla Mamba2 models with the same architecture as the nested submodels. Table 2 shows the exact configurations for each model.

**MatMamba-LM scales as well as Mamba2:** In Figure 6, we see that MatMamba-LM models scale with training tokens as well as Mamba2 models for the largest granularity. In Figure 7, we also see that for all granularities, the final trained models of every granularity scale as well as the baseline model trained with the same architecture. Furthermore, we observe that the validation loss in Figure 6 for every nested granularity is at a similar distance (usually a delta of 0.4 in val loss) between the largest model ($m_i = d_{model}$) and the smallest model ($m_i = d_{model}/8$), with consistent gaps for the intermediate models. These scaling trends offer very promising evidence that a single nested MatMamba-LM model can be used in a variety of deployments instead of training 4 separate models independently. We also report performance on a number of downstream LM eval tasks for all granularities of each MatMamba-LM model (along with baselines trained with the same architecture) in Tables 5, 6, 7, 8.

In Figure 7, we show results for adaptive inference using Mix'n'Match on all 4 MatMamba-LM variants. We see a smooth interpolation between the $d_{model}/2$ and $d_{model}$ granularities (e.g. between $d_{model}/8$ and $d_{model}/4$). However, for the lower granularities, even though the explicitly optimized granularities scale as well as expected, the Mix'n'Match models that have not been explicitly trained suffer a slight performance degradation. We observed that during earlier stages of training, the Mix'n'Match trends for all granularities were exactly on the performance-compute

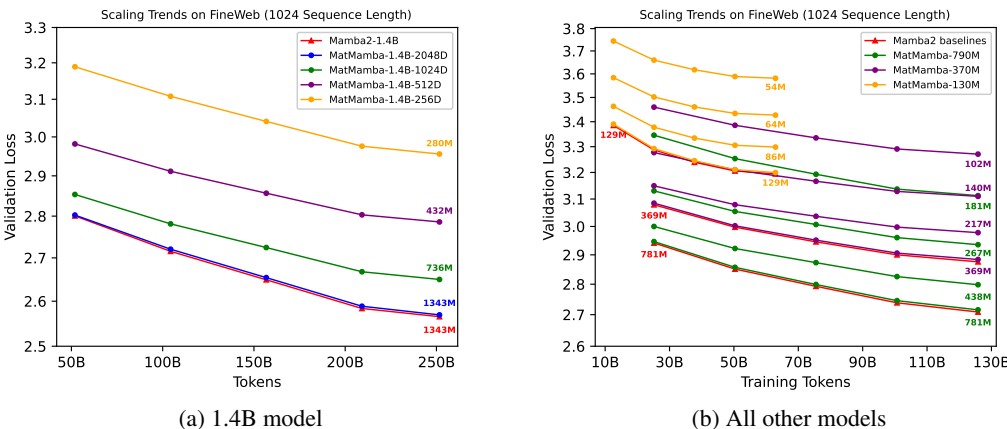

(a) 1.4B model                    (b) All other models

Figure 6: MatMamba-LM scales as well as explictly optimized Mamba2 baselines across all model and training scales all while providing accurate sub-models on the go.

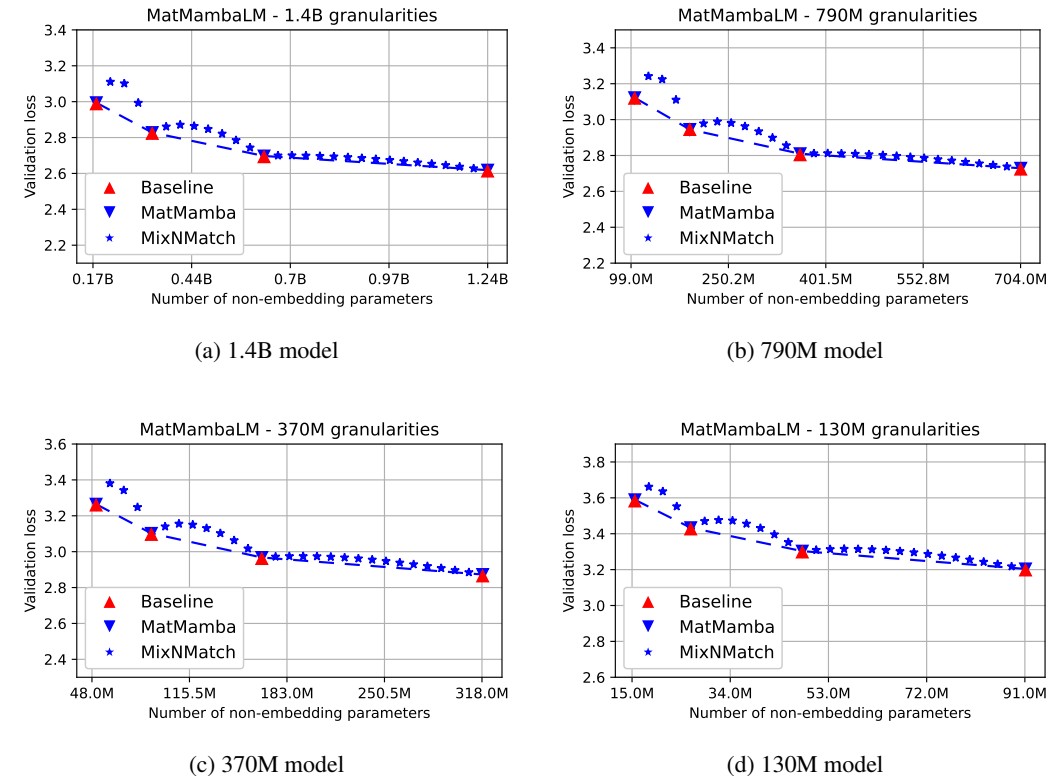

Figure 7: Validation loss for the language modelling task across model sizes showing MatMamba-LM is accurate as a Mamba2 baseline at explictely optimized granualrities, while enabling pareto optimal submodels through Mix'n'Match.

curve. However, towards the later stages, the explicitly optimized granularities improve faster than the Mix'n'Match granularities (almost like anchor points). There are mechanisms that can potentially fix this: like a self-distillation loss with the output of the largest submodel, training with more than $g = 4$ granularities, or the surrogate model structure used in Flextron (Cai et al., 2024b), that should make the Mix'n'Match trend smooth. However, this requires more rigorous understanding, and we leave deeper exploration to future work.

## 5  CONCLUSIONS

In this work, we presented MatMamba, which is a way to impose a nested Matryoshka structure on a Mamba2 state space model. It brings together the best of both Mamba-style models (faster inference times, especially for longer sequences) and Matryoshka-style learning. A single MatMamba model contains hundreds of nested and accurate submodels that can be flexibly extracted for inference. MatMamba-Vision and MatMamba-LM models match the performance and accuracy of the independently trained Mamba2 baselines. MatMamba models allow us to choose a desired performance-compute tradeoff, all while being a single Matryoshka-style model instead of multiple different models for specific scenarios. This enables interesting use cases like speculative decoding using a smaller draft model and a larger verifier model, input-adaptive submodel selection, and hybrid cloud-edge inference with the same model based on available compute.

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

# A APPENDIX

```python
# Example MatMamba parameters
d_model = 1024
expand = 2
headdim = 64
d_state = 128
d_inner = expand * d_model
n_heads = d_inner // headdim

# Learnable parameters, their shapes:
w_z       # (d_inner, d_model)
w_x       # (d_inner, d_model)
w_B       # (d_state, d_model)
w_C       # (d_state, d_model)
w_dt      # (n_heads, d_model)
D         # (n_heads)
A         # (n_heads)
w_conv_x  # (d_inner, 1, 4)
w_conv_BC # (2 * d_state, 1, 4)
w_out     # (d_model, d_inner)

def matmamba_layer(x_in, mat_dims):
    '''
    Arguments:
        x_in: (batch, seq_len, d_model)
        mat_dims: how many matryoshka dims to select in this block
    Returns:
        y: (batch, seq_len, d_model)
    '''
    mat_d_inner = expand * mat_dims
    mat_n_heads = mat_d_inner // headdim
    assert mat_d_inner % headdim == 0

    # Matryoshka structure on dims of W_z and W_x, and number of heads of W_dt
    w_in_proj = torch.cat(
        [w_z[:mat_d_inner, :], w_x[:mat_d_inner, :], w_B, w_C, w_dt[:mat_n_heads, :]],
        dim=0
    )

    zxbcdt = F.linear(x_in, w_in_proj)
    z, xBC, dt = torch.split(zxbcdt, [mat_d_inner, mat_d_inner + 2*d_state, mat_n_heads], dim=-1)

    # Matryoshka structure on W_conv_x based on mat_dims
    w_conv = torch.cat([w_conv_x[:mat_d_inner], w_conv_BC])
    xBC = F.conv1d(xBC, w_conv, groups=mat_d_inner + 2 * d_state)
    x, B, C = torch.split(xBC, [mat_d_inner, d_state, d_state], dim=-1)

    # Matryoshka structure on number of heads in dt, A, and D
    y = mamba_chunk_scan_combined(x, dt[:mat_n_heads], A[:mat_n_heads], B, C, D[:mat_n_heads])

    y = rmsnorm(y * F.silu(z), w_norm)

    # Matryoshka structure on dims of W_out
    y = F.linear(y, w_out_proj[:, :mat_d_inner])

    return y
```

Listing 1: Pytorch-style pseudocode for a MatMamba block

Table 3: Training Configuration for ImageNet runs

| Procedure | MatMamba-Vision 135M | 35M |
|---|---|---|
| Model Dim. | 1024 | 512 |
| Layers | 20 | 20 |
| Batch Size | 4096 | 8192 |
| Training Steps | 249,600 | 124,800 |
| Optimizer | AdamW | AdamW |
| LR | 0.005 | 0.005 |
| LR Decay | Cosine | Cosine |
| Weight decay | 0.1 | 0.1 |
| Warmup steps | 10,000 | 10,000 |
| Label smoothing eps. | 0.1 | 0.1 |
| Dropout | 0.1 | 0.1 |
| Stochastic depth | 0.1 | 0.1 |
| Repeated Aug | Yes | Yes |
| Gradient clip | 1.0 | 1.0 |
| Horizontal flip | Yes | Yes |
| Random Resized Crop | Yes | Yes |
| RandAugment | (2,9) | (2,9) |
| MixUp Alpha | 0.8 | 0.8 |
| CutMix Alpha | 1.0 | 1.0 |
| RandomErase prob. | 0.3 | 0.3 |
| ColorJitter | 0.3 | 0.3 |
| Test crop ratio | 1.0 | 1.0 |

Table 4: Learnable parameters (without biases) in a MatMamba layer, with example parameter reduction from a Mamba2 layer for $d_{model} = 1024$, $d_{head} = 32$, $d_{inner} = 2 \times 1024 = 2048$ (expand factor 2), $d_{state} = 128$, $m_i = 512$, and $h_i = 16$ (half of the original dimensions and half of original heads being used inside the model).

| Parameter | Mamba Shape | MatMamba Shape | Reduction Fraction |
|---|---|---|---|
| $W_z$ | $d_{inner} \times d_{model}$ $2048 \times 1024$ $2,097,152$ | $(2 \times m_i) \times d_{model}$ $(2 \times 512) \times 1024$ $1,048,576$ | 0.5x |
| $W_x$ | $d_{inner} \times d_{model}$ $2048 \times 1024$ $2,097,152$ | $(2 \times m_i) \times d_{model}$ $(2 \times 512) \times 1024$ $1,048,576$ | 0.5x |
| $W_B$ | $d_{state} \times d_{model}$ $128 \times 1024$ $131,072$ | $d_{state} \times d_{model}$ $128 \times 1024$ $131,072$ | 1x |
| $W_C$ | $d_{state} \times d_{model}$ $128 \times 1024$ $131,072$ | $d_{state} \times d_{model}$ $128 \times 1024$ $131,072$ | 1x |
| $W_{dt}$ | $n_{heads} \times d_{model}$ $32 \times 1024$ $32,768$ | $h_i \times d_{model}$ $16 \times 512$ $16,384$ | 0.5x |
| $D$ | $n_{heads}$ $32$ | $h_i$ $16$ | 0.5x |
| $A$ | $n_{heads}$ $32$ | $h_i$ $16$ | 0.5x |
| $W_{conv_x}$ | $d_{inner} \times 1 \times 4$ $2048 \times 1 \times 4$ $8,192$ | $2 \times m_i \times 1 \times 4$ $(2 \times 512) \times 1 \times 4$ $4,096$ | 0.5x |
| $W_{conv_{BC}}$ | $(2 \times d_{state}) \times 1 \times 4$ $(2 \times 128) \times 1 \times 4$ $256$ | $(2 \times d_{state}) \times 1 \times 4$ $(2 \times 128) \times 1 \times 4$ $256$ | 1x |
| $W_{out}$ | $d_{model} \times d_{inner}$ $1024 \times 2048$ $2,097,152$ | $d_{model} \times (2 \times m_i)$ $1024 \times (2 \times 512)$ $1,048,576$ | 0.5x |
| Total | $6,594,880$ | $3,428,640$ | 0.519x |

Table 5: Downstream LM Eval results for baseline and MatMamba-LM on 1.4B granularities

| Downstream Task | 256-D (d_model/8) | | 512-D (d_model/4) | | 1024-D (d_model/2) | | 2048-D (d_model) | |
|---|---|---|---|---|---|---|---|---|
| | Baseline | MatMamba | Baseline | MatMamba | Baseline | MatMamba | Baseline | MatMamba |
| LAMBADA | 36.48 | 36.74 | 43.17 | 43 | 50.24 | 50.05 | 53.77 | 53.7 |
| Hellaswag | 33.89 | 33.77 | 38.17 | 38.05 | 42.24 | 42.43 | 45.17 | 45.56 |
| WinoGrande | 50.59 | 50.67 | 55.01 | 54.54 | 56.21 | 56.12 | 58.75 | 58.64 |
| PIQA | 67.94 | 68.01 | 70.35 | 70.46 | 73.11 | 73.07 | 74.58 | 74.54 |
| ARC-E | 50.02 | 49.54 | 54.11 | 53.41 | 58.77 | 58.75 | 62.67 | 62.63 |
| ARC-C | 19.67 | 19.45 | 21.86 | 21.84 | 24.73 | 24.66 | 28.2 | 28.16 |
| OpenBookQA | 18.6 | 18.4 | 21.2 | 21.2 | 22 | 21.8 | 24.2 | 24.2 |
| TriviaQA (EM) | 0.81 | 0.74 | 3.38 | 3.31 | 7.1 | 6.98 | 9.33 | 9.31 |
| GSM8k | 0.71 | 2.12 | 1.33 | 1.36 | 1.39 | 1.21 | 1.79 | 1.67 |
| MMLU | 23.49 | 22.97 | 24.77 | 24.41 | 24.39 | 24.43 | 23.49 | 23.84 |
| ANLI-R1 | 33.58 | 31.7 | 33.4 | 31.5 | 33.27 | 33.4 | 33.71 | 33.5 |
| ANLI-R2 | 34.11 | 34.2 | 34.24 | 34.6 | 34.28 | 34.6 | 35.2 | 35 |
| ANLI-R3 | 35.12 | 35.58 | 35.17 | 34.75 | 35.34 | 35.33 | 35.27 | 33.67 |

Table 6: Downstream LM Eval results for baseline and MatMamba-LM on 790M granularities

| Downstream Task | 192-D (d_model/8) | | 384-D (d_model/4) | | 768-D (d_model/2) | | 1536-D (d_model) | |
|---|---|---|---|---|---|---|---|---|
| | Baseline | MatMamba | Baseline | MatMamba | Baseline | MatMamba | Baseline | MatMamba |
| LAMBADA | 31.34 | 31.32 | 37.71 | 37.75 | 42.4 | 42.42 | 46.59 | 46.52 |
| Hellaswag | 31.28 | 31.11 | 35.01 | 34.38 | 37.74 | 37.67 | 40.39 | 40.37 |
| WinoGrande | 50.25 | 51.38 | 51.38 | 51.38 | 51.74 | 51.7 | 54.51 | 54.54 |
| PIQA | 68.2 | 68.17 | 69.41 | 69.37 | 71.61 | 71.55 | 73.14 | 73.12 |
| ARC-E | 46.21 | 46.09 | 50.04 | 49.92 | 53.01 | 52.82 | 55.2 | 55.05 |
| ARC-C | 18.79 | 18.77 | 20.77 | 20.73 | 21.79 | 21.76 | 22.33 | 22.27 |
| OpenBookQA | 17.8 | 17.6 | 20.2 | 20 | 20.6 | 20.6 | 21.4 | 21.2 |
| TriviaQA (EM) | 0.41 | 0.37 | 0.48 | 0.45 | 2.11 | 2.08 | 3.77 | 3.64 |
| GSM8k | 1.47 | 1.9 | 1.77 | 1.74 | 1.84 | 1.82 | 1.96 | 1.97 |
| MMLU | 23.97 | 22.98 | 24.41 | 23.49 | 24.17 | 23.87 | 24.69 | 25.04 |
| ANLI-R1 | 33.41 | 32.6 | 33.58 | 29.2 | 33.71 | 32.2 | 33.17 | 32.3 |
| ANLI-R2 | 34.28 | 34.3 | 34.24 | 34.7 | 35.11 | 35.2 | 35.17 | 33.6 |
| ANLI-R3 | 35.26 | 36.25 | 35.28 | 34.75 | 34.91 | 35.08 | 35.02 | 34.58 |

Table 7: Downstream LM Eval results for baseline and MatMamba-LM on 370M granularities

| Downstream Task | 128-D (d_model/8) | | 256-D (d_model/4) | | 512-D (d_model/2) | | 1024-D (d_model) | |
|---|---|---|---|---|---|---|---|---|
| | Baseline | MatMamba | Baseline | MatMamba | Baseline | MatMamba | Baseline | MatMamba |
| LAMBADA | 27.09 | 27.09 | 32.02 | 31.98 | 37.57 | 37.55 | 42.11 | 42.05 |
| Hellaswag | 29.21 | 29.46 | 31.28 | 31.3 | 33.77 | 33.81 | 36.29 | 36.39 |
| WinoGrande | 50.77 | 51.14 | 50.74 | 50.67 | 51.22 | 51.38 | 51.68 | 51.14 |
| PIQA | 65.21 | 65.23 | 67.41 | 67.36 | 68.24 | 68.28 | 70.49 | 70.51 |
| ARC-E | 44.02 | 43.94 | 47.02 | 46.8 | 48.3 | 48.32 | 50.79 | 50.76 |
| ARC-C | 19.05 | 19.03 | 19.2 | 19.11 | 20.5 | 20.48 | 21.18 | 21.16 |
| OpenBookQA | 15.6 | 15.6 | 17.8 | 17.4 | 19.8 | 19.2 | 19.8 | 18.2 |
| TriviaQA (EM) | 0.21 | 0.2 | 0.4 | 0.42 | 0.62 | 0.61 | 1.34 | 1.32 |
| GSM8k | 1.13 | 1.59 | 1.31 | 1.29 | 1.62 | 1.59 | 1.68 | 1.59 |
| MMLU | 24.28 | 23 | 24.77 | 22.99 | 24.53 | 23.09 | 24.77 | 22.97 |
| ANLI-R1 | 33.78 | 31.9 | 33.14 | 34.3 | 33.92 | 32.7 | 33.22 | 33.7 |
| ANLI-R2 | 35.27 | 33.9 | 33.29 | 33.8 | 35.01 | 33.7 | 34.97 | 33.2 |
| ANLI-R3 | 35.78 | 34.75 | 35.12 | 35.17 | 35.89 | 33.83 | 35.58 | 34.67 |

Table 8: Downstream LM Eval results for baseline and MatMamba-LM on 130M granularities

| Downstream Task | 96-D (d_model/8) | | 192-D (d_model/4) | | 384-D (d_model/2) | | 768-D (d_model) | |
|---|---|---|---|---|---|---|---|---|
| | Baseline | MatMamba | Baseline | MatMamba | Baseline | MatMamba | Baseline | MatMamba |
| LAMBADA | 19.97 | 20.01 | 23.27 | 23.31 | 26.47 | 26.49 | 29.38 | 29.32 |
| Hellaswag | 27.63 | 27.61 | 28.34 | 28.44 | 29.38 | 29.48 | 30.11 | 30.32 |
| WinoGrande | 50.11 | 50.75 | 52.24 | 52.33 | 52.22 | 52.01 | 52.24 | 52.33 |
| PIQA | 61.93 | 61.97 | 62.31 | 62.3 | 65.97 | 64.91 | 66.52 | 66.49 |
| ARC-E | 40.87 | 40.82 | 42.31 | 42.26 | 43.67 | 43.6 | 45.12 | 45.08 |
| ARC-C | 17.31 | 17.32 | 17.43 | 17.41 | 17.85 | 17.83 | 18.89 | 18.86 |
| OpenBookQA | 12.8 | 12.4 | 14.2 | 14.4 | 15.8 | 15.6 | 16.2 | 15 |
| TriviaQA (EM) | 0.11 | 0.09 | 0.15 | 0.16 | 0.25 | 0.26 | 0.45 | 0.46 |
| GSM8k | 1.07 | 1.21 | 1.24 | 1.29 | 1.51 | 1.52 | 1.55 | 1.06 |
| MMLU | 24.11 | 22.97 | 24.39 | 22.9 | 24.92 | 22.97 | 24.33 | 22.95 |
| ANLI-R1 | 33.88 | 32.5 | 33.88 | 31.1 | 33.62 | 31.1 | 34.11 | 34.2 |
| ANLI-R2 | 34.67 | 32.8 | 34.77 | 33.1 | 34.59 | 34.6 | 34.89 | 34.4 |
| ANLI-R3 | 35.78 | 36.17 | 35.28 | 33.25 | 35.64 | 36.25 | 35.78 | 36.08 |

