# OpenReview forum: "MatMamba: A Matryoshka State Space Model"
_ICLR.cc/2025/Conference — Submitted to ICLR 2025_

### Official Review · Reviewer_vNu2 · 2024-10-26

**Soundness:** 3
**Presentation:** 3
**Contribution:** 2
**Rating:** 6
**Confidence:** 3

**Summary:**

The paper presents MatMamba the application of Matryoshka techniques (Kusupati 2022) to models based on the Mamba2 architecture.
The authors apply the Matroyshka technique to most layers, leading to a close-to linear downscaling capability.
They can show that jointly learning one model at four granularities matches the performance of models trained at the single scales.
For vision experiments, the Mix'n'Match technique using layer dimensionalities in between granularities matches/surpasses an interpolated
performance, whereas in Language Modeling there is a degradation using this technique. The presented technique binds representations at
different model scales, making potential for e.g. more efficient speculative decoding techniques.

**Strengths:**

The authors show that Matryoshka-style model tying/training can be applied to Linear Transformers/Matrix SSMs/Fast-Weight Programmers.
They show results on how the Mix'n'Match interpolation degrades performance across modalities.

**Weaknesses:**

It is a combination of existing techniques.

**Questions:**

Can you give a citation / relevant prior work for the claim in L.456f ?
There is a typo in L.522.
Do you think that the different Mix'n'Match behavior for language and image modeling is intrinsic to the modality or might be specific to MatMamba?

---

> ### Author Response · Authors · 2024-11-22
> **Official Response to Reviewer vNu2**
>
> We thank the reviewer for their response. We appreciate that they recognize the close-to-linear downscaling capability of MatMamba when compared to other methods. We address the reviewer’s questions and concerns below:
>
> > It is a combination of existing techniques.
>
> We would like to highlight the rapidly growing adoption of the Mamba (and other linear attention / efficient recurrent style models) in both research and production environments. We believe that combining the adaptivity of Matryoshka-style learning and the efficiency of state space models (SSMs) like Mamba is a novel contribution.
>
> > Can you give a citation / relevant prior work for the claim in L.456f ?
>
> The scaling laws series of works (e.g. Kaplan et al. 2020 and related follow ups) offer evidence to support the claim. However, we agree that presenting downstream eval results gives a clearer picture of the model performance. We have amended line 456 to no longer contain the original claim, and we also present detailed results for 13 downstream tasks (LAMBADA, Hellaswag, PIQA, ARC etc) using the standard lm-evaluation-harness settings. The results show a similar trend to the validation loss in which performance improves with model size for certain tasks (e.g. PIQA, LAMBADA etc.).
>
> > There is a typo in L.522.
>
> Fixed
>
> > Do you think that the different Mix'n'Match behavior for language and image modeling is intrinsic to the modality or might be specific to MatMamba?
>
> We think that this is unclear and is up for further exploration. The language models we trained are larger than the image models (and the ImageNet dataset is also balanced/easier to learn similar representations for 1000 classes, whereas the LM vocab size is approx. 50k and needs a larger number of training steps). We observed that for the earlier stages of training, the Mix’n’Match LM curves also lie exactly on the performance-compute curve. However, towards the later stages, the explicitly optimized granularities improve faster than the Mix’n’Match granularities (almost like anchor points). This needs more rigorous understanding.
>
> As to whether this behavior is specific to MatMamba, we point the reader to figure 2 (a) of the MatFormer paper (https://arxiv.org/abs/2310.07707) which also has a similar trend (albeit slightly less pronounced). This suggests that the behavior is likely not unique to MatMamba.

---

> > ### Author Response · Authors · 2024-11-27
> >
> > As the deadline for potential revisions is due in a few hours, we wanted to check back if the reviewer had any other comments or questions after our rebuttal.
> >
> > Looking forwards to hearing from you and happy to answer any further questions.

---

> > > ### Comment · Reviewer_vNu2 · 2024-12-02
> > >
> > > Thank you for your reply and insights! No more to add from my side.

---

### Official Review · Reviewer_jnCr · 2024-11-04

**Soundness:** 2
**Presentation:** 4
**Contribution:** 3
**Rating:** 6
**Confidence:** 4

**Summary:**

This paper presents MatMamba, which introduces a nested Matryoshka structure on a Mamba2 state space model. The resulting elastic network architecture can be used to generate multiple smaller sub-networks at deployment time without any additional training, similar to prior work in this area such as Matformer and Flextron. The training process involves gradient accumulation on 4 subnetworks with a single backward pass for parameter updates. MatMamba is evaluated on vision and language tasks with elastic network sizes ranging from 35M to 1.4B.

**Strengths:**

* The paper is very well-written, with the main ideas and results presented clearly. It was a pleasure to read.
* Elastic networks have shown significant promise for Transformer networks, since they can be used to generate tailored networks for specific deployment scenarios at no additional training cost (once elastic training is complete). The paper makes a contribution in this relevant and promising direction.
* Evaluation and results across vision and language looks strong overall.

**Weaknesses:**

* There doesn’t seem to be a systematic way of selecting smaller sub-networks given a deployment constraint. For a given parameter or latency budget, should all dimensions be proportionally reduced? How is this proportion decided?
* Novelty: this work appears to be a straightforward application of Matryoshka principles to the SSM domain; apart from the obvious changes needed for SSMs, the training algorithm remains nearly identical to the Matformer work.
* It’s not clear if the scaling trends continue beyond the ~1.5B parameter mark. Do the authors have any insights here? I understand that training larger networks can be quite resource-intensive, but at least a discussion on expected scaling trends would be useful.
* I don’t agree with the claim made on line 456. Downstream task performance must be reported for MatMamba-LM using a standard framework like LM-Eval-Harness. While validation loss is a good starting point, performance on a variety of downstream tasks can provide a more complete picture of LLM performance (which is why most work dealing with LLMs reports these numbers).

**Questions:**

* Why is g set to 4? Have you explored other values? How does the method scale when g is lower/higher?
* How is MatMamba's long context performance? SSM-based architectures can perform poorly on long-context tasks, and I’d be curious to see if MatMamba maintains this trend, or perhaps improves it. MatMamba’s superior performance at higher resolutions in the vision domain provide somewhat of an indication here.

---

> ### Author Response · Authors · 2024-11-22
> **Official Response to Reviewer jnCr (1/2)**
>
> We thank the reviewer for their response. We are glad that they enjoyed reading the paper! We address the reviewer’s questions and concerns below:
>
> > There doesn’t seem to be a systematic way of selecting smaller sub-networks given a deployment constraint. For a given parameter or latency budget, should all dimensions be proportionally reduced? How is this proportion decided?
>
> This is a fantastic question. We appreciate the reviewer’s attention to detail :)
>
> For a given parameter budget, one could do a number of things including reducing all dimensions proportionally, reducing some dimensions a lot but keeping others the same, or various combinations of these that fit the particular deployment hardware or other constraints.
>
> In the work, we empirically found that for image models, a pyramid/inverted pyramid type of dimension selection (e.g. 512-D in first 12 layers, then 1024-D in next 12 layers; or the other way round) worked better for ImageNet models, whereas proportionally reducing all dimensions worked better for language models (e.g. for 50% of parameter count, we choose all dimensions to be 1024 in a base 2048-D model).
>
> The convenient thing is that we can choose all of these dynamically on-the-fly, so just a few quick validation set passes with a number of different combinations can help tune this for the task at hand. We are studying more systematic (instead of just empirical) methods for this process in future work.
>
> > Novelty: this work appears to be a straightforward application of Matryoshka principles to the SSM domain; apart from the obvious changes needed for SSMs, the training algorithm remains nearly identical to the Matformer work.
>
> We believe that combining the adaptivity of Matryoshka-style learning and the efficiency of state space models (SSMs) like Mamba is a novel contribution (2 very different types of algorithmic efficiency being demonstrated to work well together). Also, MatMamba has several advantages over MatFormer (e.g. >=95% of parameter count being in the projection layers, which makes tensor parallelism efficient and easier without needing FlashAttention etc, or that we do the Matryoshka process over the whole network instead of just the MLP layers which leads to a larger parameter reduction).
>
> > It’s not clear if the scaling trends continue beyond the ~1.5B parameter mark. Do the authors have any insights here? I understand that training larger networks can be quite resource-intensive, but at least a discussion on expected scaling trends would be useful.
>
> We expect the scaling trends to continue. The strongest intuition we have here is: given that there exists a duality between Mamba2 and Transformers (see detailed discussion in the Mamba2 paper https://arxiv.org/pdf/2405.21060), and that scaling trends continue on Transformers, we would expect them to also continue on Mamba2. We will add more detailed discussion on (expected) scaling trends.
>
> > I don’t agree with the claim made on line 456. Downstream task performance must be reported for MatMamba-LM using a standard framework like LM-Eval-Harness. While validation loss is a good starting point, performance on a variety of downstream tasks can provide a more complete picture of LLM performance (which is why most work dealing with LLMs reports these numbers).
>
> Thank you for the valuable suggestion (and for the specific pointer to lm-evaluation-harness). We have now reported results for 13 downstream tasks (LAMBADA, Hellaswag, PIQA, ARC etc) using the standard lm-evaluation-harness settings. We have also amended line 456 to no longer contain the original claim, which could be confusing to readers.

---

> ### Author Response · Authors · 2024-11-22
> **Official Response to Reviewer jnCr (2/2)**
>
> > Why is g set to 4? Have you explored other values? How does the method scale when g is lower/higher?
>
> g=4 is an empirical choice (can be adjusted as needed), although choosing D, D/2, D/4, and D/8 as the four granularities was intentional in order to keep the training cost theoretically within the ceiling of 2X FLOPs compared to just training a single model (given that N + N/2 + N/4 + … = 2N). This also made it easy to compare our results with MatFormer. In some smaller scale experiments, we did observe that choosing extremely low values like D/32 can be counterproductive because it increases the overhead while also being potentially noisy/unstable for representation learning in the most important first few dimensions.
>
> > How is MatMamba's long context performance? SSM-based architectures can perform poorly on long-context tasks, and I’d be curious to see if MatMamba maintains this trend, or perhaps improves it. MatMamba’s superior performance at higher resolutions in the vision domain provide somewhat of an indication here.
>
> Another great question! While we do not explicitly do super long-context evals in this work (beyond processing images at higher resolutions), there is strong evidence that Mamba2 works very well on long context tasks (when compared to the original Mamba or prior SSM’s that struggled with these). Please see the MQAR results in Figure 8 of the Mamba2 paper (https://arxiv.org/abs/2405.21060), on which the original Mamba (and even Attention) has much worse performance. Moreover, (Mat)Mamba models are significantly faster at longer context lengths for both training and inference. In future work, we plan to apply MatMamba to long-form video.
>
> Once again, thank you for the detailed review with several great observations which we think will help improve the work!

---

> > ### Author Response · Authors · 2024-11-27
> >
> > As the deadline for potential revisions is due in a few hours, we wanted to check back if the reviewer had any other comments or questions after our rebuttal.
> >
> > Looking forwards to hearing from you and happy to answer any further questions.

---

> > > ### Comment · Reviewer_jnCr · 2024-11-27
> > >
> > > Thank you for the prompt and detailed response. I commend the authors for collecting additional results on downstream tasks in this limited rebuttal timeframe. However, the lack of a systematic and automated way for selecting sub-networks remains a concern for me. I will maintain my current score.

---

> > > > ### Author Response · Authors · 2024-11-27
> > > >
> > > > Thanks for the commendation! We agree that sub-network selection in Mix'n'Match might seem raphazard, but in practice, this can be modelled as a constraint optimization problem during inference based on the compute requirements or even learned with additional compute. Overall, the capabiltiy of having so many strong pareto optimal models, with a small inference time search cost is still worth it for downstream deployments. We hope to make the search process more systematic in the near future.

---

### Official Review · Reviewer_JmGL · 2024-11-04

**Soundness:** 3
**Presentation:** 3
**Contribution:** 3
**Rating:** 5
**Confidence:** 3

**Summary:**

This paper introduces MatMamba, a novel approach to State Space Models (SSMs) that combines Mamba model with Matryoshka-style learning. The proposed model embeds nested sub-models within a single super-architecture, allowing for dynamic inference across different granularities without the need for separate training. This setup facilitates scalable and efficient deployment in both vision and language models by dynamically adjusting model sizes based on compute availability. Through extensive testing on tasks such as image classification and language modeling, MatMamba demonstrates comparable or superior performance to baseline Mamba2 models while maintaining flexibility and efficiency.

**Strengths:**

(1) Adaptive Inference Resource: MatMamba allows the extraction of multiple nested submodels from one single model, enabling versatile deployment options, from edge devices to cloud settings, without the need for retraining. By using the Mix’n’Match approach, MatMamba optimizes for various performance-compute trade-offs, allowing resource allocation adjustments based on current needs.
(2) Scalability and Performance: The model exhibits scalability on par with transformers and baseline Mamba2 models in both image and language tasks. The author provides multiple model sizes ranging from 30M parameters to 1.4B parameters, which demonstrates scalability.
(3) The nested structure maintains a consistent metric space, ensuring that different granularities yield compatible outputs, which is beneficial for downstream applications like retrieval.

**Weaknesses:**

(1) Dependency on Explicitly Trained Granularities for Optimal Performance: While MatMamba’s Mix’n’Match approach offers flexibility, untrained granularities do not perform as effectively as explicitly optimized ones, showing degradation in performance and accuracy.
(2) Limited Exploration of Self-Distillation Techniques: The paper mentions that self-distillation or other techniques could further enhance interpolation accuracy in untrained granularities, suggesting room for improvement in the training setup.
(3) Limited Evaluation on Language Tasks: The author trained a large model up to 1.4B parameters, but did not conduct more experiments on text-generation benchmarks used to evaluate LLMs. I reckon that it would be better if the author could run the proposed MatMamba through datasets like MMLU, HellaSwag, GSM8K, ARC and etc.

**Questions:**

See the weaknesses

---

> ### Author Response · Authors · 2024-11-22
> **Official Response to Reviewer JmGL**
>
> We thank the reviewer for their response. We are glad that they appreciated the adaptive inference, scalability, and performance of MatMamba, and that they noted the positive effect that the consistent metric space has on tasks like retrieval. We address the reviewer’s concerns and questions below:
>
> > (1) Dependency on Explicitly Trained Granularities for Optimal Performance: While MatMamba’s Mix’n’Match approach offers flexibility, untrained granularities do not perform as effectively as explicitly optimized ones, showing degradation in performance and accuracy.
>
> While we acknowledge the slight degradation of validation loss for the untrained granularities in MatMamba-LM, we would like to highlight that in the MatMamba-Vision models, the untrained granularities are not only at par, but often exceed the expected performance on the performance-compute curve. The retrieval results also clearly indicate that the models preserve the metric space, which is an extremely useful property (even in cases where the validation loss slightly suffers in the untrained granularities).
>
> Additionally, Mix’n’Match is very useful in helping pick near pareto-optimal models for free. Often, this is not feasible without additional training. If someone does use a Mix’n’Match model from an untrained granularity that is slightly lesser performant, even that can offer an extremely strong starting point for finetuning with a small number of steps.
>
> > (2) Limited Exploration of Self-Distillation Techniques: The paper mentions that self-distillation or other techniques could further enhance interpolation accuracy in untrained granularities, suggesting room for improvement in the training setup.
>
> Self-distillation techniques (e.g. DistillBERT) are complementary to Matryoshka-style training (e.g. one could use the output of the largest model to train the smaller ones). However, these necessitate additional training runs and need more compute. We will explore these ideas in-depth in future work.
>
> > (3) Limited Evaluation on Language Tasks: The author trained a large model up to 1.4B parameters, but did not conduct more experiments on text-generation benchmarks used to evaluate LLMs. I reckon that it would be better if the author could run the proposed MatMamba through datasets like MMLU, HellaSwag, GSM8K, ARC and etc.
>
> Thank you for the valuable suggestion. We have now added results for downstream LM evals in Tables 5,6,7, and 8. The results show a similar trend to the validation loss in which performance improves with model size for certain tasks (e.g. PIQA, LAMBADA etc.). Note however, that downstream results at this scale are fairly noisy, whereas validations loss/scaling laws are more predictive of model performance.

---

> > ### Author Response · Authors · 2024-11-27
> >
> > As the deadline for potential revisions is due in a few hours, we wanted to check back if the reviewer had any other comments or questions after our rebuttal.
> >
> > Looking forwards to hearing from you and happy to answer any further questions.

---

### Official Review · Reviewer_Rpty · 2024-11-04

**Soundness:** 4
**Presentation:** 4
**Contribution:** 3
**Rating:** 6
**Confidence:** 4

**Summary:**

This work extends Matryoshka Representation Learning to Mamba2, a representative architecture for state-space models, proposing MatMamba. MatMamba introduces a training pipeline that generates numerous sub-models, each offering different performance-efficiency trade-offs, within a single training run. The authors analyze the effectiveness and scalability of the proposed method.

**Strengths:**

- This work validates the scalability of Matryoshka Representation Learning by demonstrating its applicability to state-space models (SSMs). It considers both Mamba-vision and Mamba-language models, confirming effectiveness across both tasks.
- The methodology and problem setting are well-established and well-motivated.

**Weaknesses:**

- For the MatMamba-LM model family, only evaluation loss is reported. The submission would be strengthened by including evaluations on downstream tasks as well.

**Questions:**

- What is the main difference you observed between elastic inference on Matformer and MatMamba during the experiment? Since MatMamba essentially follows the approach used in Matformer, I'm curious if any unique challenges were encountered specific to the Mamba architecture.

---

> ### Author Response · Authors · 2024-11-22
> **Official Response to Reviewer Rpty**
>
> We thank the reviewer for their response. We are glad that the reviewer agrees that MatMamba validates the scalability/effectiveness of Matryoshka learning when applied to SSMs, and that the problem and our methodology are well motivated. We address the reviewers concerns and questions below:
>
> > For the MatMamba-LM model family, only evaluation loss is reported. The submission would be strengthened by including evaluations on downstream tasks as well.
>
> We appreciate the reviewer’s suggestion. We have now included results for MatMamba-LM on a number of downstream tasks (in Tables 5,6,7,8 in the Appendix). The results show a similar trend to the validation loss in which performance improves with model size for certain tasks (e.g. PIQA, LAMBADA etc.).
>
> > What is the main difference you observed between elastic inference on Matformer and MatMamba during the experiment? Since MatMamba essentially follows the approach used in Matformer, I'm curious if any unique challenges were encountered specific to the Mamba architecture.
>
> A crucial difference between MatFormer and MatMamba is that MatFormer only does nesting on the MLP portion of the Transformer block, whereas MatMamba does the nesting on all applicable parts of the model. Thus, the parameter count (and theoretical inference speed assuming optimal hardware utilization) reduces linearly in MatMamba with nesting, whereas in MatFormer the attention parts stay constant. Follow-up works like Flextron have extended the idea to attention layers as well, but contain additional design elements (e.g. a surrogate model and a latency-based loss) which we did not study in the context of Mamba-like models in this work.
>
> Another interesting property of the Mamba2 model is that >=95% of the typical parameter count lies in the input and output projection layers alone (whereas in a transformer it is split 40-60 between attention and MLP portions). This helps us use tensor parallelism easily and effectively.
>
> While we did not encounter any other major challenges, we are eager to build upon this work and further optimize the nesting mechanism, which is an open problem.

---

> > ### Author Response · Authors · 2024-11-27
> >
> > As the deadline for potential revisions is due in a few hours, we wanted to check back if the reviewer had any other comments or questions after our rebuttal.
> >
> > Looking forwards to hearing from you and happy to answer any further questions.

---

### Author Response · Authors · 2024-11-22
**Updated Submission in Response to Feedback**

We thank the reviewers for their valuable feedback. We appreciate the positive comments about MatMamba’s flexibility, scalability, and efficiency. We have now updated the submission in response to the comments.

The reviewers unanimously agreed that adding downstream LM evals would strengthen the paper. We now report results on 13 benchmarks (LAMBADA, PIQA, ARC, Hellaswag etc) for all granularities of all MatMamba-LM models in Tables 5,6,7, and 8 in the Appendix. We have also amended the confusing claim we earlier made in line 456 that two reviewers pointed out.

We are happy to discuss any further questions, clarifications, or feedback about the work.

---

### Meta-Review · Area_Chair_zCzi · 2024-12-13

**Metareview:**

The submission presents MatMamba, a novel application of Matryoshka-style representation learning to state-space models (SSMs), enabling flexible nested submodels within a single architecture. While the paper provides a thorough exposition of the method and demonstrates its scalability and efficiency across language and vision tasks, it struggles to establish sufficient novelty beyond prior work, such as Matformer and Flextron. The approach appears to be a relatively straightforward extension of existing techniques, and the core contributions are incremental. The dependence on explicit training for specific granularities and limited exploration of broader applications or benchmarks further weakens its impact. Additionally, while some improvements over baseline models are observed, the paper fails to provide insights into challenges unique to MatMamba or its broader implications for SSM-based architectures.

Despite the strong presentation and interesting results, the lack of a systematic approach for submodel selection, as well as underexplored areas such as self-distillation and long-context performance, limits the paper’s contribution to the field. Furthermore, the paper would benefit from additional evaluations on more diverse and challenging language modeling benchmarks to substantiate its claims.

**Additional Comments On Reviewer Discussion:**

During the rebuttal, reviewers raised concerns about the limited novelty of the work, dependency on explicitly trained granularities for optimal performance, lack of a systematic approach for submodel selection, and insufficient evaluation on downstream language tasks. The authors addressed these by adding results for 13 benchmarks in the appendix, amending unclear claims, and elaborating on MatMamba's advantages over similar works like Matformer. However, they acknowledged the lack of systematic submodel selection and deferred further exploration to future work. While the added evaluations and clarifications strengthened the paper, the reliance on existing techniques and incremental contributions ultimately led to the decision to recommend rejection, as these limitations outweighed the positive aspects.

---

### Decision · Program_Chairs · 2025-01-22

Reject